# Scalable Generation of Mesenchymal Stem Cells and Adipocytes from Human Pluripotent Stem Cells

**DOI:** 10.3390/cells9030710

**Published:** 2020-03-13

**Authors:** Manale Karam, Ihab Younis, Noor R. Elareer, Sara Nasser, Essam M. Abdelalim

**Affiliations:** 1Diabetes Research Center, Qatar Biomedical Research Institute (QBRI), Hamad Bin Khalifa University (HBKU), Qatar Foundation (QF), PO Box, Doha 34110, Qatar; manale.karam@ac.bioscience.com (M.K.); NElareer@hbku.edu.qa (N.R.E.); SNassar@hbku.edu.qa (S.N.); 2Biological Sciences Program, Carnegie Mellon University in Qatar, Qatar Foundation, Education City, Doha 24866, Qatar; iyounis@andrew.cmu.edu; 3College of Health and Life Sciences, Hamad Bin Khalifa University (HBKU), Qatar Foundation (QF), Education City, Doha 34110, Qatar

**Keywords:** retinoic acid, hPSCs, MSCs, apoptosis, proliferation, RNA-Seq, Adipogenic differentiation

## Abstract

Human pluripotent stem cells (hPSCs) can provide unlimited supply for mesenchymal stem cells (MSCs) and adipocytes that can be used for therapeutic applications. Here we developed a simple and highly efficient all-*trans*-retinoic acid (RA)-based method for generating an off-the-shelf and scalable number of human pluripotent stem cell (hPSC)-derived MSCs with enhanced adipogenic potential. We showed that short exposure of multiple hPSC lines (hESCs/hiPSCs) to 10 μM RA dramatically enhances embryoid body (EB) formation through regulation of genes activating signaling pathways associated with cell proliferation, survival and adhesion, among others. Disruption of cell adhesion induced the subsequent differentiation of the highly expanded RA-derived EB-forming cells into a pure population of multipotent MSCs (up to 1542-fold increase in comparison to RA-untreated counterparts). Interestingly, the RA-derived MSCs displayed enhanced differentiation potential into adipocytes. Thus, these findings present a novel RA-based approach for providing an unlimited source of MSCs and adipocytes that can be used for regenerative medicine, drug screening and disease modeling applications.

## 1. Introduction

Mesenchymal stem cells (MSCs) have been demonstrated to be a promising option for cellular therapies given their curative properties of immunomodulation, trophic support and homing, and differentiation into specific cells of a damaged tissue, as well as their poor immunogenicity allowing allogenic transplantation without strong immunosuppressants [1]. However, difficulties in obtaining therapeutic numbers of MSCs with appropriate differentiation capabilities have hampered the use of these multipotent cells in clinics. MSCs can be obtained from various adult (bone marrow and adipose tissue) and perinatal (cord blood and placenta) sources. However, MSCs compose a negligible fraction of cells within in vivo tissues and need to be subjected to the process of in vitro expansion [2]. Isolation of MSCs from adult sources involves invasive and often painful procedures with possible donor site morbidity [2]. Furthermore, whatever the source, the isolated MSCs show heterogeneity in their proliferation and differentiation capabilities, which are further reduced during in vitro expansion [3].The variations in the properties of isolated MSCs are due to the differences in the nature of their niche, donor age and their isolation, as well as culturing methods.

Alternatively, human pluripotent stem cells (hPSCs) can provide an unlimited source of MSCs [4,5,6,7], which exhibit characteristic MSC properties. Those include adherence and fibroblast-like morphology in standard culture conditions, expression of the MSC markers CD44, CD73, CD90 and CD105, lack of the hematopoietic markers CD14, CD19, CD34 and CD45, and the ability to further differentiate into the three mesodermal lineages (chondrocytes, adipocytes and osteocytes) following a definite stimulation [8,9].

In addition to their direct therapeutic potential, owing to their ability to differentiate into adipocytes, MSCs may represent a valuable in vitro cell-based model to study adipogenesis, obesity and related diseases. However, the efficiency of the currently available protocols for adipocytic differentiation from hPSC-derived MSCs is modest and sometimes involves genetic manipulations, which can potentially modify the activity of other genes and the cellular phenotype [10,11,12,13].

All-*trans*-retinoic acid (RA) is a metabolite of vitamin A that mediates several crucial physiological processes [14,15]. RA acts by binding to its receptor (RAR), which heterodimers with retinoid X receptor (RXR) and binds to retinoic acid response elements in DNA, thereby inducing the transcription of a wide array of specific target genes involved in cell growth and differentiation [15]. Although RA has been widely described as an inducer of cell differentiation, RA can antagonize cell differentiation and induce stemness depending on the concentration, time of exposure and cell type [16]. For example, while high RA concentrations induce the differentiation process of hPSCs into pancreatic and neuronal cell lineages [17,18], short exposure of PSCs to low concentrations of RA sustains pluripotency and suppresses differentiation [17]. Surprisingly, comparably low concentrations of RA have been shown to promote differentiation of adipocytes from mouse ESCs [19]; however, the overall efficiency remains modest and the mode of control of these antagonistic effects of RA is not clear. Furthermore, whether RA affects MSC differentiation is unknown.

In this report, we present a highly efficient RA-based method for generating, from multiple hPSC lines (hESCs/hiPSCs), an off-the-shelf and scalable number of MSCs and adipocytes. Our data showed that short treatment of hPSC-derived embryoid bodies (EBs) with 10 μM RA dramatically enhanced EB-forming cell proliferation, survival and self-renewal through the activation of several signaling pathways. The subsequent differentiation of these highly expanded cells into MSCs could be induced by disruption of cellular interaction. Interestingly, the generated MSCs displayed enhanced differentiation potential into adipocytes that can be used for disease modeling and cellular therapy.

## 2. Materials and Methods

### 2.1. Culture of Human Pluripotent Stem Cells

hESC lines (H1 and H9) were obtained from WiCell Research Institute (Maddison, WI, USA). Several hiPSC lines were generated from healthy individuals in our lab. Those hiPSC lines were extensively characterized as described in our recently published article [20]. hESCs/hiPSCs were cultured in StemFlex media (Thermo Fisher Scientific, Waltham, MA, USA) and were passaged using ReLeSR (Stem Cell Technologies, Vancouver, BC, Canada) or EDTA. The cells were treated with 10 μM Y-27632 (Stemgent, USA) for 24 h after passaging.

### 2.2. Differentiation of Human Pluripotent Stem Cells into Mesenchymal Stem Cells (MSCs)

To differentiate hPSCs into MSCs, we modified a previously published protocol [11]. hPSC colonies were dissociated with ReLeSR (Stem Cell Technologies) into small clumps and cultured in differentiation media containing 10 μM Y-27632 (Stemgent, Boston, MA, USA) on ultra-low attachment plates to form embryoid bodies (EBs). The differentiation media consisted of low glucose DMEM (Thermo Fisher scientific) supplemented with 15% fetal bovine serum (FBS) and 1% penicillin/streptomycin. At day 2 of differentiation, the medium was replaced with fresh media supplemented with 0, 0.1, or 10 µM retinoic acid (RA) (Sigma-Aldrich, St. Louis, MO, USA) (Figure 1A), while at day 4, the media were replaced with fresh media supplemented with 0 or 0.1 μM RA (Figure 1A). At day 6 of differentiation, the media were replaced with differentiation medium without RA. At day 7, EB differentiation into MSCs was induced by EB plating onto gelatin/matrigel-coated plates in the differentiation medium, which was replaced every two days (Figure 1A). At day 12, the medium was replaced with differentiation medium containing 2.5 ng/mL bFGF (i.e., MSC growth medium) and refreshed every two days. The MSCs were passaged using trypsin/EDTA when they reached 80–90% confluency. The MSC identity and phenotype of the differentiated cells were checked by flow cytometry analysis of MSC markers (CD44, CD73, CD90, CD105, CD271) at different time points. The MSCs were cryopreserved in FBS containing 10% DMSO.

### 2.3. Differentiation of MSCs into Adipocytes

For adipogenic differentiation, MSCs were seeded at 2.5 × 10^4^ cells/cm^2^ density and cultured in MSC growth medium. When the MSCs reached 100% confluency, the MSC growth medium was replaced with adipogenic differentiation medium. Two different methods adapted from previously reported protocols were used for adipogenic differentiation [13,21] with some modifications. Those protocols allow the generation of adipocytes without genetic manipulation of the cells. In protocol 1 (Pr1), the adipogenic differentiation medium consisted of knockout DMEM-F12 (Thermo Fisher Scientific) supplemented with 10% knockout serum replacement (KSR), 1% glutamax, 1% penicillin/streptomycin, 0.5 mM 3-isobutyl-1-methylxanthine (IBMX), 0.25 µM dexamethasone, 1 µg/mL insulin, 0.2 mM indomethacin and 1 µM pioglitazone (all from Sigma-Aldrich) [13]. While in protocol 2 (Pr2), the media consisted of MEM-alpha (Thermo Fisher Scientific) supplemented with 10% FBS, 1% penicillin/streptomycin, 100 µg/mL IBMX, 1 µM dexamethasone, 0.2 U/mL insulin, 100 μM indomethacin and 10 μM Roziglitazone [21]. The adipogenic medium was changed every two days. The adipocytic differentiation was assessed by examining lipid accumulation using Oil Red O and BODIPY staining and adipogenesis marker expression (FABP4, PPAR γ and adiponectin) by immunocytochemistry and/or flow cytometry after 10–14 days.

### 2.4. Oil Red O Staining and Quantification

For Oil Red O staining, the cells were fixed with 4% paraformaldehyde (PFA) for 1h at room temperature. After fixation, two washes with dH_2_O and one wash with 60% isopropanol, the cells were allowed to dry before staining with filtered 0.4% Oil Red O solution in 60% isopropanol for 1 h at room temperature. The cells were then washed with dH_2_O to remove unbound dye. Then, lipid droplets were visualized and photographed under light microscope.

To quantify Oil Red O staining, the cells were allowed to dry then the dye was eluted in 100% isopropanol by incubation for 10 min at room temperature on a shaker. After pipetting up and down several times, 75 μL was transferred to two wells of a flat-bottom 96-well plate. Then, the absorbance was measured at 492 nm, with 100% isopropanol used as blank. Undifferentiated MSCs stained with Oil Red O as described above were used as control. Sample absorbance was corrected by subtracting the absorbance obtained for blank and the absorbance obtained for undifferentiated MSCs.

### 2.5. Differentiation of MSCs into Osteocytes and Chondrocytes

To induce osteogenic differentiation, confluent hPSC-derived MSCs were cultured in MEMα medium supplemented with 10% FBS, 100 nM dexamethasone and 200 µM ascorbic acid. The medium was changed twice a week for 21 days [22]. The osteogenic differentiation was assessed by examining the deposition of calcium using Alizarin Red staining. Therefore, the differentiated cells were fixed with 4% PFA for 30 min at room temperature, washed twice with dH_2_O and stained for 5 min with 2% Alizarin red solution pH 4.2. After wash with dH_2_O, Ca^2+^ deposits were visualized and imaged under light microscope.

To induce chondrogenic differentiation, hPSC-derived MSCs were cultured at 1.25 × 10^6^ cells/mL in chondrogenic medium in 96-well V-bottom plate (200 µL/well). The chondrogenic medium consisted of high glucose DMEM supplemented with 1% Insulin-Transferrin-Selenium, 1.25 mg/mL serum albumin, 37.5 µg/mL ascorbate-2-phosphate, 10^−7^ M dexamethasone, 1% nonessential amino acids and 10 ng/mL TGF-β1. The medium was changed every 2 days for 21 days [22]. The chondrogenic differentiation was assessed by examining the accumulation of glycosaminoglycan (GAG) using alcian blue staining. Therefore, chondrocyte spheres were collected, fixed with 4% PFA for 30 min, washed with phosphate-buffered saline (PBS) containing 0.5% Triton and embedded to freeze in Optimal Cutting Temperature (O.C.T.) Compound at −80 °C prior to sectioning on a cryostat (Leica). Then, chondrocyte sections were stained for 30 min at room temperature with 1% alcian blue solution prepared in 0.1 M HCL. After rinsing with 0.1 M HCl, PBS and H2O, the accumulation of GAGs was visualized and imaged under light microscope (Olympus, Tokyo, Japan).

### 2.6. Immunocytochemistry

Adipocytes derived from hPSC-derived MSCs were washed with PBS (Thermo Fisher Scientific) and fixed in 4% PFA in 0.1 M PBS (pH 7.4) (Santa Cruz Biotechnology, Dallas, TX, USA) for 30 min. The cells were permeabilized for 15–20 min with 0.5 % Triton X-100 (Sigma, St. Louis, MO, USA) in PBS (PBST) and blocked overnight with 6% bovine serum albumin (BSA) in PBST. The cells were incubated at 4 ºC overnight with primary antibodies. After washing the cells three times with tris-buffered saline with 0.5 % Tween 20 (TBST), the cells were incubated with Alexa Fluor secondary antibodies (Thermo Fisher Scientific). Nuclei were counterstained with Hoechst 33342 (1 µg/mL) (Thermo Fisher Scientific) and the plates were assessed using Olympus IX53 inverted fluorescence microscopy (Tokyo, Japan). For lipid droplet fluorescent staining, BODIPY™ 493/503 (Molecular Probes, D3922) was used at 5 µM. The details of primary and secondary antibodies are listed in Appendix A. 

### 2.7. Flow Cytometry

For intracellular target staining, the cells were collected using trypsin/EDTA (25%), washed with PBS, then fixed with 4% PFA for 15 min at RT. After fixation, the cells were permeabilized and blocked with 6% BSA in PBS containing 0.5% saponin for at least 1 h at RT. Then, the cells were stained with primary antibodies (1:100 dilution) prepared in incubation buffer (PBS containing 1% BSA and 0.5% saponin) for 30 min at RT. After two washes with incubation buffer, the cells were incubated with Alexa Fluor secondary antibodies (1:500 dilution) for 30 min at RT. Then, the cells were washed twice with incubation buffer and resuspended in PBS. For cell surface target staining, the cells were collected using trypsin/EDTA (25%), washed with cold PBS and incubated with fluorochrome-conjugated antibodies (1:100 dilution) for 20 min at 4 °C. After incubation, the cells were washed twice with cold PBS and resuspended in PBS. For both types of staining, data was acquired on a BD Accuri™ C6 flow cytometer and analyzed using FlowJo software. Isotype controls were used as negative controls to help differentiate non-specific background signal from specific antibody signal. The details of primary, secondary and isotype control antibodies are listed in Appendix A.

### 2.8. Cell Proliferation Assays

Effect of RA treatment of the proliferation of EB-forming cells was examined using BrdU incorporation and CFSE cell proliferation assays. 

BrdU incorporation assay was performed as previously described [23] with some modifications. Briefly, the cells were incubated without or with BrdU (1:100; ThermoFisher Scientific, Waltham, MA, USA) for 19 h in the differentiation media. After incubation, the cells were dissociated using trypsin/EDTA (25%), washed with PBS and then fixed with cold 70% ethanol at −20 °C overnight. Fixed cells were denatured with 2 M HCl containing 0.5% Triton and neutralized by 0.1 M sodium borate. Then, the cells were stained with Alexa Fluor 488-conjugated anti-BrdU antibody (1:100; ThermoFisher Scientific) for 3 h at RT. BrdU incorporation in the cells was analyzed using BD Accuri™ C6 flow cytometer (BD Biosciences, Franklin Lakes, NJ, USA). The results were processed using FlowJo.

CFSE cell proliferation assay was performed on H9-derived EB forming cells untreated or treated with RA (10 µM treatment condition). Therefore, on day 0 (D0) of differentiation, just before EB induction, H9-hESCs were stained with 5 μM CellTraceTM CFSE dye (ThermoFisher Scientific) by incubation for 20 min at 37 °C. After incubation, the cells were collected in five times the original staining volume of pre-warmed differentiation medium (containing 15% FBS) and incubated for 5 min to remove any free dye remaining in the solution. The cells were then pelleted and resuspended in fresh pre-warmed differentiation medium and cultured as per the PSC to MSC differentiation protocol described above. CFSE staining in the cells within the EBs was assessed overtime at days 1, 4 and 7 of differentiation by flow cytometry. The percentages of proliferating cells (percentage of cells with decreased CFSE stain relative to day 1 of differentiation) were determined by FlowJo software (version 10.5.3). The median fluorescence intensities (ΔMFIs) (representative of the overall cell proliferation rate) were calculated by subtracting MFI (unstained) from MFI (stained).

### 2.9. Cell Viability Assay

EB-forming cells were dissociated using trypsin/EDTA (25%) and 1 × 10^5^ cells were resuspended in 500 µL of 1 × Annexin V-binding buffer. Then, 5 µL Annexin V-FITC (Abcam, # ab14085) and 5 µL propidium iodide (PI) were added and incubated for 5 min before analysis by flow cytometry to determine the percentages of viable (Annexin V-negative and PI-negative), early apoptotic (Annexin V-positive and PI-negative) and late-apoptotic/necrotic (Annexin V-positive and PI-positive) cells.

### 2.10. RNA Sequencing and Data Analysis

RNA was extracted from at least two biological replicates for each sample of cells at days 3 and 5 of differentiation using Direct-zol RNA extraction kit (Zymo Research). mRNA was captured using NEBNext (Poly A) mRNA magnetic isolation kit (NEB, E7490) according to the manufacturer’s instructions. RNA-seq libraries were prepared using NEBNext ultra directional RNA library prep kit (NEB, E7420L) and sequenced on an Illumina Hiseq 4000 system. The RNA-seq reads were stored in fastq files and aligned to the UCSC hg38 reference genome using the STAR aligner with default parameters [24]. The average length of mapped reads for the samples was between 150–151 nucleotides.

Only uniquely mapped reads (stored in bam files) were used for further processing. Differentially expressed genes were identified using Cuffdiff [25] by computing fragment per kilobase per million mapped reads (FPKM) and taking into account local and global differences in the distribution of mapped read. Cuffdiff compares samples with varying replicate numbers and outputs *p*-values (for statistical significance) and fold changes of the observed difference between samples. For differentially expressed genes only genes with FPKM > 0.5 were considered and cutoffs of *p*-value < 0.05 and more than 1.5-fold change were set. Visualization (Venn diagrams, volcano plots and heatmaps) of RNA-seq data was done with the combined replicates using R.

### 2.11. Real Time PCR Analysis

Real time RT-PCR analysis was performed using GoTaq qPCR Master Mix (Promega) as previously described [20]. The primer details are listed in Appendix A. 

### 2.12. Enrichment Analysis

Enrichment analysis was performed with DAVID. The upregulated and downregulated genes were subjected to GO analysis using DAVID Bioinformatics Resources 6.8. 

### 2.13. Statistical analysis

Data are expressed as mean ± SD or SEM (as indicated). Statistical analysis was performed using Student’s t test or one-way ANOVA followed by Dunnett’s test when indicated. A *p* value < 0.05 was considered significant.

## 3. Results

### 3.1. Short-Term RA Treatment Enhances Embryoid Body (EB) Formation from hPSCs by Inducing Both Cell Proliferation and Survival 

To test whether RA can enhance the generation of MSCs from hPSCs, we first checked the effect of this retinoid on the formation of EBs, i.e., intermediates in the hPSC to MSC differentiation. Therefore, hPSCs were dissociated into small aggregates and cultured in suspension to allow EB formation. The forming EBs were transiently treated with 0.1 or 10 µM RA from day 2 to day 5 as indicated in Figure 1A. Then, their number was assessed on day 7. EB formation could be observed the next day (D1) of hPSC clump transfer to low-adhesion dishes. The number of the EBs was progressively reduced from a day to another (D2–D7) (data not shown) and their size was heterogeneous (Figure 1B; Appendix A). Importantly, the hPSCs (hESCs/hiPSCs) that were treated with RA allowed the generation of more EBs than those that were not treated (Figure 1B, Appendix A). Treatment with 0.1 µM and 10 µM RA significantly increased the number of hESC-derived EBs, by around 3.8- and 7.8-fold, respectively, in comparison to the untreated condition (0 µM) (Figure 1B). These results suggest that a 4-day treatment with RA enhances the formation and/or maintenance of EBs in a dose-dependent manner. Similar to hESCs, short-term treatment with 10 μM RA enhances the generation of EBs from hiPSCs (Appendix A). In fact, all five RA-treated hiPSC clones formed significantly more EBs than their RA-untreated counterparts; fold increases ranged from 8.9 to 55.56 depending on the hiPSC clone (mean fold increase of 33.64) (Appendix A).

RA might enhance EB formation by regulating cell proliferation and/or survival. The effect of RA treatment on cell proliferation was examined by two independent assays, BrdU incorporation (Figure 1C) and CFSE cell proliferation (Figure 1D) assays. The number of BrdU-positive cells increased from 23.6% to 37% when the H9 hESC-derived EBs were treated with 10 µM RA, reflecting an increase in cell proliferation (Figure 1C). Consistently, H9 hESC-derived EB treatment with 10 µM RA significantly increased the percentage of cells exhibiting decrease in CFSE staining (i.e., % of proliferating cells in EBs) and the CFSE median fluorescence intensity (i.e., the proliferation rate of EB-forming cells). Thus, EB treatment with RA enhances the proliferation of EB-forming cells. Next, to assess the effect of RA on the survival of EB-forming cells, Annexin V/propidium iodide staining was performed on 2-day-old EBs after 4 h of treatment with RA (Figure 1E). The results showed that RA treatment significantly increased the proportion of viable cells (Annexin V-/Propidium iodide-) in the H9 hESC-derived EBs from 35.7% (without RA) to 52.4% (with RA). Taken together, our data indicate that short-term treatment with 10 µM RA enhances EB formation from hPSCs by inducing both cell proliferation and survival.

Next, we thought to check whether concentrations of RA higher than 10 µM would allow further increase in EB generation. Therefore, at day 2 of differentiation, EBs derived from H9-hESCs were treated with 10 µM, 20 µM, 50 µM and 100 µM RA. Two days after RA treatment (day 4 of differentiation), we morphologically noticed a reduction in the number of EBs treated with 50 µM, which was more dramatic than those treated with 100 µM (Appendix A). This reduction in EB number was more evident at day 5 of differentiation (Appendix A). At day 7 of differentiation (the day of plating), the EBs were counted from two independent experiments. The results showed significant and highly significant reduction in the number of EBs treated with 20 µM and 50 µM, respectively, in comparison to those treated with 10 µM (Appendix A). Furthermore, we only detected a few numbers of EBs treated with 100 µM, indicating severe cell death in EBs treated with high concentrations of RA (Appendix A). Thus, 10 µM RA, which generates the highest EB number, was used for their subsequent differentiation into MSCs.

### 3.2. RA-Mediated Inhibition of PSC-Derived EB Differentiation into MSCs Can Be Relieved by Cell Dissociation

After plating on matrigel-coated dishes, RA-untreated H9 hESC-derived EBs lost their compact shape and started spreading as quickly as the next day (D8). At D12, fibroblast-like cells grew and migrated out of the EBs (Figure 2A, 0 µM RA). On the other hand, RA-treated EBs maintained their compact shape after plating and failed to differentiate into fibroblast-like cells (Figure 3A, 0.1 and 10 µM RA).

Since cell–cell and cell–extracellular matrix adhesions play an important role in regulating survival, proliferation, self-renewal and differentiation of hPSCs [26,27,28], we checked whether the disruption of cell adhesion may induce the differentiation of the plated RA-treated EBs into fibroblast-like cells under MSC growth medium. Therefore, after 5 days of plating on matrigel-coated dishes in MSC differentiation medium, RA-treated H9 hESC-derived EBs were dissociated (on D12 of differentiation) and replated on matrigel-coated dishes in MSC growth medium (Figure 2B). After 4 and 8 days of culture (D16 and D20 of differentiation, respectively), the cellular morphology (Figure 2C) and the expression of the specific MSC marker CD73 (Figure 2D) were assessed by imaging and flow cytometry, respectively. At D16 of differentiation, most RA-untreated EBs were spontaneously resolved by cell migration and differentiation into CD73-expressing fibroblast-like cells without any cell dissociation (Figure 2C,D; RA 0 µM, P0/D16). Contrariwise, RA-treated EBs still maintained their compact shape and few cells grew out of these EBs after 16 days of differentiation in the absence of cell dissociation (Figure 2C; RA 10 µM, P0/D16). These cells did not display clear elongated (fibroblast-like) shapes, maintained strong cell-to-cell adhesion and did not significantly express the CD73 marker (Figure 3D, RA 10 µM, P0/D16). However, the dissociation and replating of these cells on D12 induced their differentiation into fibroblast-like cells very efficiently (Figure 2C; RA 10 μM, P1/D16), most of which (around 86%) expressed the MSC marker CD73 on day 16 of differentiation (Figure 2D; RA 10 μM, P1/D16). Further culturing allowed full differentiation of the dissociated RA-treated EBs into MSCs (99.9% CD73-positive cells on day 20 of differentiation) (Figure 2C,D; RA 10 μM, P2/D20). These results suggest that cell dissociation induces the differentiation of RA-treated EBs into MSCs.

Following the above results, cell dissociation and replating was always performed on day 12 of hPSC differentiation into MSCs in the next experiments (as in Figure 1A).

### 3.3. Short-Term EB Treatment with RA Strongly Enhances MSC Differentiation Yield

Since our results showed that short-term treatment with RA significantly enhances the formation of EBs that display a strong potential to differentiate into MSCs, we subsequently tested the effect of RA on the yield of MSC differentiation. Therefore, EB formation, maturation and differentiation into MSCs were performed as described in Figure 1A, using the H9 hESCs (Figure 3), H1 hESCs (Appendix A) and different hiPSC clones derived from three different healthy donors (Appendix A) as the sources of hPSCs. At D20 of differentiation, the morphology, number and phenotype of the generated MSCs were compared. The results show that in all three RA treatment conditions, elongated MSC-like cells were formed from H9-hESCs (Figure 3A). The number of these cells was significantly increased by around 3.8-fold when the EBs were treated with 0.1 μM RA and dramatically increased by around 18-fold when the EBs were treated with 10 μM RA (Figure 3B). More than 99% of these MSC-like cells expressed the MSC markers CD44, CD73 and CD90 no matter the RA treatment condition (Figure 3C,D). On the other hand, 77.7% of MSCs obtained from the differentiation of RA-untreated EBs expressed the CD105 marker, which was strongly decreased to 42% and 35% by EB treatment with 0.1 and 10 μM RA, respectively (Figure 3C,D). Whatever the EB treatment condition (with or without RA), a small population (4–29%) of the generated MSCs expressed the bone marrow MSC marker CD271, with no significant difference between the three conditions (Figure 3C,D). Similarly to H9-hESC-derived EBs, short-term treatment of H1-hESC-derived EBs with 10 μM RA generated phenotypically comparable MSCs (CD44^high^, CD73^high^, CD90^high^, CD105^low^, CD271^low/−^) with strongly enhanced yield by around 18-fold in comparison to their RA-untreated counterparts (Appendix A).

Treatment of hiPSC-derived EBs with 10 μM RA induced a more dramatic increase in the MSC yield, ranging from a 11.2- to 1542-fold increase for five independently performed differentiations using three different hiPSC clones generated from three different healthy donors (Appendix A). 

The phenotype of the MSCs generated by the differentiation of RA-treated hiPSC-derived EBs was comparable to the phenotype of MSCs generated from RA-treated H1/H9-hECS-derived EBs; i.e., CD44^high^, CD73^high^, CD90^high^, CD105^low^, CD271^low/−^ (Appendix A).

Of note, all of the differentiated MSCs lacked the expression of the hematopoietic markers (CD45, CD34, CD19 and CD14) (Appendix A).

Taken together, these results indicate that short-term treatment of hPSC-derived EBs with 10 μM RA strongly enhances the generation of MSCs that display characteristic elongated fibroblast-like morphology and high CD44, CD73 and CD90 MSC marker expression but low/negative CD105 and CD271 marker expression.

### 3.4. Long-Term Culture, Freezing and Thawing of the hPSC-Derived MSCs

Long-term culture of the MSCs derived from RA (10 μM)-treated EBs showed that these cells can be maintained and amplified in culture for at least 34 days (D50 of differentiation) while maintaining MSC phenotypes comparable to those observed on D20 of differentiation (Figure 4A).

To check whether the MSCs derived from RA (10 μM)-treated EBs could be cryopreserved, those were frozen on D20 of differentiation and stored at −150 °C. After 2 months, the frozen MSCs were thawed, their viability was checked using trypan blue and then they were cultured on gelatin-coated dishes in MSC growth medium (Figure 4B). The thawed MSCs were 100% viable (data not shown), reached confluency after 24 h of thawing and were passaged every ~3 days at 1/3 or 1/4 dilutions. These cells maintained their fibroblast-like shape and their initial expression levels of MSC markers for at least two weeks (Figure 4C,D). Thus, these results indicate that the RA (10 μM)-treated EBs can be cryopreserved for later downstream applications.

### 3.5. Confirmation of the Multipotency of the hPSC-derived MSCs by Trilineage Differentiation

MSCs derived from EBs treated or not with 10 μM RA and frozen on day 25 of differentiation (passage 4) were thawed, maintained in culture for around one week (2 passages) and then differentiated into the three mesenchymal lineages (adipocytes, chondrocytes and osteocytes). The results showed that both types of MSCs (obtained under 0 and 10 μM RA conditions) were capable of differentiating into adipocytes (Figure 5A), chondrocytes (Appendix A) and osteocytes (Appendix A). These findings confirm the multipotency of the generated MSCs that was maintained after cryopreservation and at least 7 passages.

### 3.6. MSCs Derived from RA-treated EBs Display an Increased Adipocytic Differentiation Potential

In the present study, our purpose was to produce, from hPSCs, an off-the-shelf and scalable source of MSCs with the ultimate intent of using them for subsequent differentiation into adipocytes to study adipocyte-related disorders. Therefore, MSCs obtained from H9-hESC-derived EBs treated or not with 10 µM RA were differentiated into adipocytes using two different previously reported protocols. After 10 days of culture in the adipocytic differentiation media, the cells were stained for lipid accumulation (Oil Red O and BODIPY) and for adipogenesis molecular markers (FABP4, PPARγ and Adiponectin).

Both types of MSCs, i.e., derived from RA-treated and untreated EBs, were able to generate adipocytes as confirmed by lipid droplet staining (Oil Red O and BODIPY) and adipogenesis marker expression (FABP4, PPARγ and adiponectin) (Figure 5). However, the multipotent cells obtained from RA-treated EBs, surprisingly, generated more adipocytes than the MSCs obtained from RA-untreated EBs with both protocols of adipocytic differentiation used. In fact, following their culture in adipogenesis media, MSCs derived from RA-treated EBs generated more cells accumulating lipid droplets than MSCs derived from RA-untreated EBs, with around a 3-fold increase in intracellular lipid accumulation (Oil Red O staining) (Figure 5A). Furthermore, MSCs derived from RA-treated EBs generated around 77.4% (protocol 1) or 48.5% (protocol 2) FABP4-positive cells, while MSCs derived from RA-untreated EBs generated only around 57.6% and 22.5%, respectively (Figure 5B). Thus, RA-treated EBs allow the generation of MSCs with 2-fold increased potential to differentiate into FABP4-positive cells, which was further evidenced by immunostaining and fluorescence microscopy (Figure 5C). Consistently, more adipocytic markers such as PPARγ (Figure 5C) and BODIPY-staining (Figure 5D) were enhanced in adipocyte-differentiated RA-derived MSCs compared to control MSCs (0 µM RA). However, no difference in the expression level of adiponectin could be observed between the two types of MSCs (Figure 5D), probably due to a problem in the specificity or application compatibility of the used antibody. Taken together, these results strongly indicate that the MSCs derived from RA-treated EBs display an enhanced potential to differentiate into adipocytic lineage.

Furthermore, we examined the ability of the MSCs derived from hiPSCs to generate adipocytes. Using the same protocol used for H9-derived MSCs, we differentiated hiPSCs-Ctr1 and hiPSCs-Ctr2 into adipocytic lineage. We found that MSCs obtained from hiPSC-derived EBs treated with 10 µM RA could efficiently differentiate into large number of adipocytes (Appendix A). Those adipocytes expressed key adipogenic markers, including FABP4, BODIPY and adiponectin (Appendix A). These results indicate the reproducibility of our protocol in hESCs and hiPSCs.

### 3.7. Transcriptome Profiling of Cells Treated with Different Concentrations of RA

To gain insight into the mechanism by which RA enhances the generation of hPSC-derived MSCs, we performed RNA-sequencing (RNA-seq) analysis on H9 derived hESC-derived EBs at days 3 and 5 of differentiation. Our analysis identified 215 upregulated and 126 downregulated genes in the EBs treated with 0.1 μM RA in comparison to untreated EBs at day 3 of differentiation, while 323 upregulated and 265 downregulated genes were identified in the EBs treated with 10 μM RA in comparison to untreated EBs at day 3 of differentiation (Log2 FC, *p* < 0.05) (Appendix A). Furthermore, at day 5 of differentiation, we identified 555 upregulated and 385 downregulated genes in the EBs treated with 0.1 μM RA in comparison to untreated EBs, while 895 upregulated and 796 downregulated genes were identified in the EBs treated with 10 μM RA in comparison to untreated EBs (Log2 FC, *p* < 0.05) (Appendix A). The differentially expressed genes (DEGs) were subjected to Gene Ontology (GO) and Kyoto Encyclopedia of Genes and Genomes (KEGG) pathway enrichment analysis to explore biological functions and pathways enriched in EBs treated with RA (Figure 6A,B). The GO categories of gene sets, which were significantly upregulated or downregulated in day 3 and day 5 in the EBs in response to 0.1 μM RA and 10 μM RA showed some common GO terms when compared with untreated EBs. However, those terms were more significant in the EBs treated with 10 μM RA (Figure 6A,B; Appendix A).

At day 3 of differentiation, the upregulated genes in the EBs treated with 10 μM RA led to several enriched GO terms of biological processes and pathways, such as pathways regulating multipotency of stem cells, Ras-associated protein-1 (RAP1), Hippo, TGF-beta, WNT and tight junction (*p* < 0.001, Figure 6A, Appendix A). The GO terms of the downregulated genes at day 3 included WNT, pluripotency of stem cells, Hippo, cGMP-PKG and axon guidance (*p* < 0.001, Figure 7). On the other hand, the upregulated genes at day 5 of differentiation led to several enriched GO terms of biological processes and pathways, such as positive regulation of cell proliferation, negative regulation of apoptosis process, pathways regulating multipotency of stem cells, cell adhesion molecules (CAMs), focal adhesion, extracellular matrix (ECM) receptor interaction, tight junction, Hippo, TGF-beta, osteogenic differentiation and mineral absorption, p53, fatty acid biosynthesis and lipid metabolism and positive regulation of epithelial mesenchymal transition (*p* < 0.001, Figure 6B,C, Appendix A). For the downregulated genes at day 5, GO terms included several pathways such as WNT, pluripotency of stem cells, Hippo, PI3K, axon guidance, RAP1, TGF-beta, adherens junctions and Notch (Figure 6B,C; Appendix A).

Of those pathways, there were several upregulated DEGs at day 5, associated with CAMs and tight junction, such as cadherin (CHD1, CHD3, CHD5), claudin genes (CLDN1, CLDN2, CLDN3, CLDN4, CLDN6, CLDN9, CLDN23), integrin subunit alpha (ITGAV) (Figure 6C) (Appendix A). Furthermore, several upregulated genes were found to be involved in focal adhesion and ECM-receptor interaction, such as SPP1, MYL7, FN1, ITGA3, ITGAV, PDGFRA, COL1A1, FLNB, FLNC, TNC, LAMA5, LAMB1, MYL12B, BCAR1, JUN, VEGFB, THBS1, SDC4, HSPG2, DAG1 and SV2A (Figure 6C; Appendix A). Furthermore, we observed a significant upregulation in genes involved in TGF-β signaling pathway, such as LEFTY2, LEFTY1, SMAD6, SMAD7, NODAL, BMP4, ID1, TGFB1, CDKN2B, GDF6, AMHR2, PITX2, BAMBI and THBS1 (Figure 6C; Appendix A). Interestingly, some genes associated with osteogenic differentiation and mineral absorption were significantly upregulated, such as FOSL1, FOSL2, MAPK12, FOSB, GAB2, TGFB1, JUN, JUNB, NFKBIA, SOCS3, SQSTM1, CYBA, SLC40A1, HMOX1, STEAP1, STEAP2, FTH1 and ATP1A1. Some fatty acid metabolism genes were significantly upregulated, such as ACOT4, ELOVL6, ELOVL1 and HACD2 (Appendix A; Appendix A).

To validate the RNA-seq results, we examined the mRNA expression levels of selected DEGs in H9-hESC-derived EBs treated with 10 µM RA compared to those treated with 0.1 µM RA and untreated EBs at day 5 of differentiation (Figure 7). The analysis showed a significant upregulation in genes involved in the positive regulation of cell proliferation (ISL1, EPCAM and SOX4) and genes involved in negative regulation of apoptosis (GATA6, KRT18 and BMP4) and the gene of cell adhesion molecules, HLA-C. Furthermore, the results showed a significant downregulation in genes involved in WNT signaling pathway, i.e., CCDN1 (cyclin D1) and WNT1, as well as those involved in cell pluripotency, OCT4 and NANOG (Figure 7). These findings confirm the RNA-seq data.

## 4. Discussion

In the present study, we developed a simple, cost-effective and highly efficient method for the generation of scalable numbers of readily available MSCs that display maintained viability, expansion capacities and multipotency following cryopreservation as well as enhanced adipogenic differentiation potential. This was achieved by a simple but revolutionary modification of the most straightforward and efficient currently available protocol of hPSC differentiation into MSCs [11]. This modification consisted of a short-term treatment of the hPSC-derived EBs with 10 μM RA.

When the differentiation was performed in the absence of RA treatment, the initial protocol allowed the generation of differentiation cultures containing ≥ 89% MSCs, as previously reported [11]. Although the purity of this population is quite high, the numbers of the obtained MSCs are rather modest at early passages and still require in vitro expansion for further applications/differentiation. However, similarly to primary MSCs, PSC-derived MSCs also exhibit reduced proliferation and differentiation capabilities following in vitro expansion [11,29]. On the other hand, starting with the same number of PSCs and for the same duration of differentiation processes, short-term treatment with 10 μM RA induced a dramatic increase in the MSC yield, ranging from 11.2 to 1542-fold depending on the hESC and hiPSC lines used, without any further in vitro expansion. In fact, short RA treatment increased proliferation and survival of EB-forming cells as demonstrated by BrdU incorporation and apoptosis assays, thereby enhancing EB formation. However, using higher concentrations of RA (above 20 μM) showed a negative effect on the formation and survival of the formed EBs, indicating that 10 μM RA is an optimal concentration for MSC generation. Consistently, RNA-seq data revealed that short RA treatment induced the expression, in the EB-forming cells, of a wide range of genes involved in the positive regulation of cell proliferation, negative regulation of apoptotic process as well as cell–cell and ECM–cell adhesions, which are also widely known as critical players in enhancing the proliferation and survival of most cell types, including PSCs [27]. Among the identified genes, many have been previously described as targets of the RA in different biological contexts. For example, *RARA* [30], *TBX3* [31], *TBX2* [32], *BAMBI* [33], *HAS2* [34], *DAB2* [30], *ID1* [35], *VEGFB* [36] and *THBS1* [37] have been identified as direct transcriptional targets of RA, while other RA-induced differentially expressed genes such as *IGF2*, *EDN1*, *FGFR4*, *PDGFRa*, *TNC*, *TGFB1*, *FOSL1*, *GATA6*, *KRT18*, *Serpin*, *BMP4*, *TFAP2A*, *CDKN1A* [30], *SMAD6* [38], *DAB2* [30], *CYR61* [39], *EGR3* [40], *SOCS3* [41] and *SQSTM1* [42] have been previously described to be indirectly regulated by RA, usually through a transcriptional intermediary. These observations further support our finding that the enhanced proliferation and survival of hPSC-EB-forming cells is a specific effect of RA [43]. Thus, in the present study, we highlight the pro-proliferative role of short RA treatment in hPSCs, in contrast to its widely recognized function as a negative regulator of cell proliferation in differentiated normal cells [44,45] as well as in cancer cells [46].

The RA-induced proliferation and survival of EB-forming cells led to a dramatic increase in the number of EBs. However, the generated EBs displayed a delayed differentiation in comparison to RA-untreated EBs, which agrees with a recent study that showed that short RA exposure suppresses hiPSC differentiation by inhibiting the Wnt canonical pathway [17]. Consistently, we found a significant downregulation of the WNT signaling pathway in RA-treated EBs in comparison to their RA-untreated counterparts, which might be involved in RA-mediated differentiation suppression. Furthermore, short RA treatment strongly increased the expression of around 50 cell adhesion genes, which are known to play a critical role in maintaining self-renewal and the pluripotent state of hPSCs. Notably, numerous cell adhesion molecule (CAM) family members were identified on the surface of hPSCs and found to directly regulate self-renewal and pluripotency [27]. Among these, *E-cadherin* (*CDH1*), which mediates intercellular interactions, plays an important role in the survival and self-renewal of hESCs and its expression is used to demarcate differentiated and undifferentiated hESCs. Upregulation of E-cadherin expression significantly increases the cloning efficiency and self-renewal capacity of hESCs [47] and its expression is decreased immediately after induction of differentiation [48,49]. Thus, the increased expression of E-cadherin and other adhesion molecules in RA-treated EBs might suggest a less differentiated state in comparison to RA-untreated EBs, which was efficiently reverted by the disruption of cell adhesion. In fact, RA-treated EB differentiation was rapidly induced by EB dissociation and cell replating, further supporting the major role of cell adhesion in the RA-induced delay in EB differentiation.

It is noteworthy that although RA induced self-renewal and suppressed differentiation of cells forming hPSC-derived EBs, RA did not seem to sustain the pluripotency of these cells, but rather block their differentiation at an early-stage precursor state. In fact, RNA-seq data showed that short RA-treatment of hPSC-derived EBs significantly decreased the expression of several pluripotency-related genes, including the core pluripotency factors, *OCT4*, *NANOG* and *SOX2*, and induced the expression of other multipotency-related genes in a dose-response manner. Taken together, these findings suggest that short treatment with RA enhances cell proliferation and survival while reducing pluripotency and inducing early differentiation of EB-forming cells, but blocks late differentiation due to increased cell adhesion. Subsequently, RA-mediated delay in EB differentiation can be reverted by disruption of cell adhesion following cell dissociation.

Short treatment with 10 μM RA followed by cell dissociation of H9-derived EBs allowed their differentiation into a pure population of MSCs, similarly to RA-untreated EBs but in much higher quantities (up to 1542-fold increase). The generated MSCs could be easily stored and maintained for viability, expansion capacities, phenotype and multipotency (efficient differentiation into the three mesenchymal lineages; adipocytes, chondrocytes, and osteocytes) following cryopreservation. Thus, short-term treatment of hPSC-derived EBs with 10 μM RA is a highly efficient method to manufacture clinically relevant amounts of MSCs that meet all criteria defining MSCs, including adherence to plastic, fibroblast-like morphology, surface expression profile and lineage commitment. However, although in the present study MSCs were not produced in accordance with Good Manufacturing Practice (GMP) guidelines, which is critical for therapeutic applications, we expect that short RA treatment of PSCs cultured in GMP-compatible conditions would yield similar large-scale expansion of functional MSCs. In fact, comparison of hPSC-derived MSCs cultured in a traditional medium supplemented with FBS (as in the case of the present study) and alternative culture conditions, which meet the requirements for clinical translation, showed that these media yield cells with very similar properties [50]. Thus, RA is a promising tool for manufacturing large amounts of clinical-grade MSCs, which is a paramount to the success of stem cell therapy using these cells.

Interestingly, the MSCs derived from RA-treated EBs displayed an enhanced potential to differentiate into adipocytic lineage. This finding is of high importance since the efficiency of currently available protocols for adipocytic differentiation from hPSC-derived MSCs is very low in the absence of genetic manipulations [10,11,12,13]. RNA-seq data showed that RA regulates several signaling pathways that play a critical role in adipogenic differentiation of MSCs, including the downregulation of the Hippo signaling pathway [51], the upregulation of extracellular matrix-receptor interaction genes [52,53] and the downregulation of the WNT pathway [54,55,56,57]. In addition to the enrichment of pro-adipogenic genes, RA allowed the generation of MSCs displaying reduced CD105 expression. Although the role of CD105 in human adipogenesis is unknown, the lack of expression of this protein from murine MSCs was found to be associated with enhanced differentiation into adipocytes in comparison to their CD105-positive counterparts [58], in agreement with our present study. Thus, RA might empower the generation of increased numbers of patient-specific adipocytes by allowing the production of their precursors (i.e., MSCs) form patient-derived iPSCs not only in high numbers but also with enhanced adipocytic prone properties, thereby providing a valuable in vitro model to study adipogenesis, obesity and related diseases.

Finally, functional studies assessing the properties of the MSCs to promote the anti-inflammation and immunosuppression in an injury model would be important to confirm the potential of RA-derived MSCs to treat inflammatory processes. This property was not assessed in the present study but preliminary data from RNA sequencing analysis on MSCs obtained from H9-hESCs without and with treatment with 10 µM RA showed that in both conditions these MSCs express some factors (such as TB4, galactin-1, TGF-β1, VEGFA, VEGFB, CCL2, PDL1, PDL2, BMP4, IL-6, HGF, TSG6, IL-10, IDO1, IDO2 as well as TNF-α and IFN-γ receptors) that are involved in MSC immunomodulatory and paracrine functions with no significant differences between the two conditions (Appendix A). Further studies and functional analyses are required to confirm the immunomodulatory function of the RA-derived MSCs.

## 5. Conclusions

In conclusion, our current study provides a novel method for the generation of large numbers of multipotent MSCs that can be expanded in culture for several passages. Those MSCs have a high capacity to be further differentiated into adipocytes. This RA-based approach can be used to generate sufficient hPSC-derived MSCs and adipocytes for in vitro disease modeling and drug screening studies.

## Figures and Tables

**Figure 1 cells-09-00710-f001:**
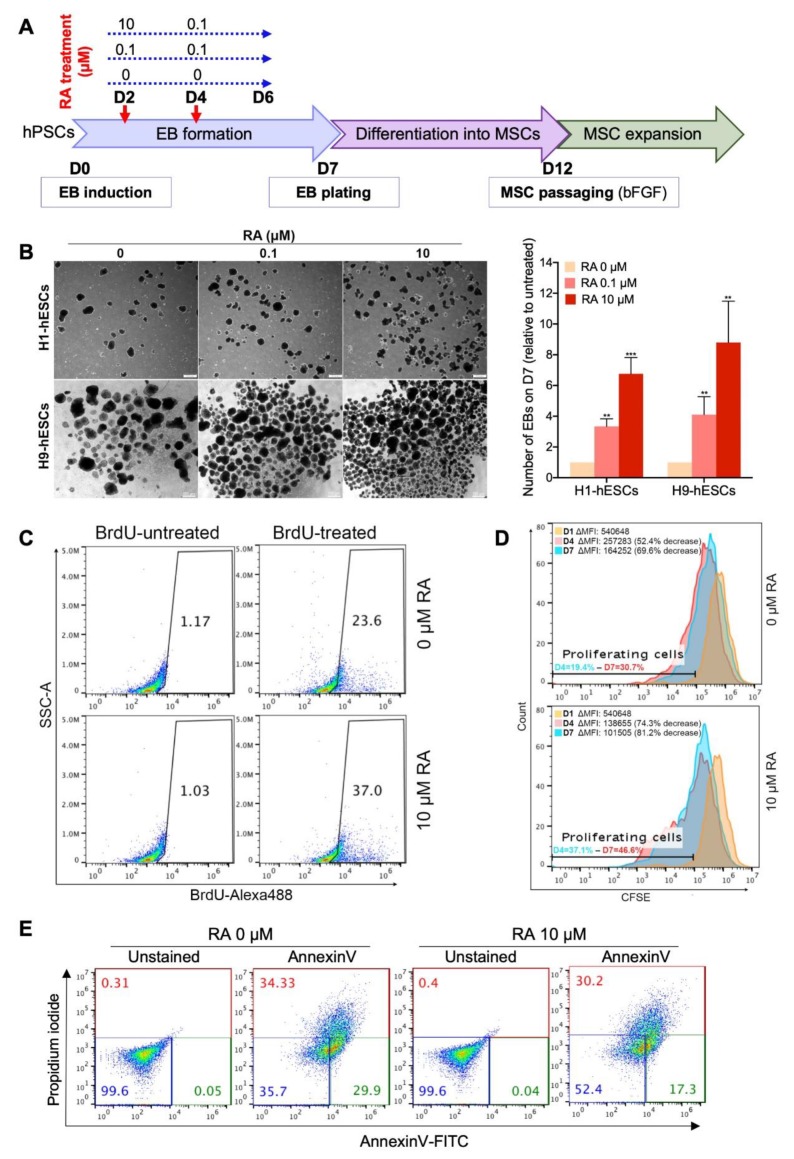
Short-term all-*trans*-retinoic acid (RA) treatment enhances embryoid body (EB) formation by inducing cell proliferation and survival. (**A**) Experimental scheme for the differentiation of human pluripotent stem cells (hPSCs) into mesenchymal stem cells (MSCs). (**B**) Representative images of H1 and H9 hESC-derived EBs showing the effect of RA concentration on EB formation (*n* = 4). The EBs were quantified at D7 of differentiation. The results presented in the bar graph are means ± SD for three independent experiments. * *p* < 0.05, ** *p* < 0.01 and *** *p* < 0.001 versus untreated EBs (0 µM) (Dunnett’s test). (**C**,**D**) Effect of RA treatment of the proliferation of EB-forming cells. (**C**) Flow cytometry analysis of BrdU incorporation during 19 h in D3-old H9-derived EBs treated or not with 10 µM RA for 6h at D2. The gates on the dot plots represent BrdU-positive cells and are representative of two independent experiments. (**D**) CFSE cell proliferation assay performed on H9-derived EB forming cells untreated (upper histogram) or treated (lower histogram) with RA (10 µM treatment condition). The H9-hESCs were stained with CFSE on D0 just before EB induction then CFSE staining decrease in proliferating cells was assessed overtime by flow cytometry. (**E**) Effect of RA treatment on the viability of EB-forming cells. H9-derived EBs were treated on day 2 with 10 µM RA for 4 h then stained using Annexin V-FITC and propidium iodide and analyzed by flow cytometry. The percentages of viable (blue), early apoptotic (green) and late-apoptotic/necrotic (red) cells are indicated on the dot plots and are representative of two independent experiments.

**Figure 2 cells-09-00710-f002:**
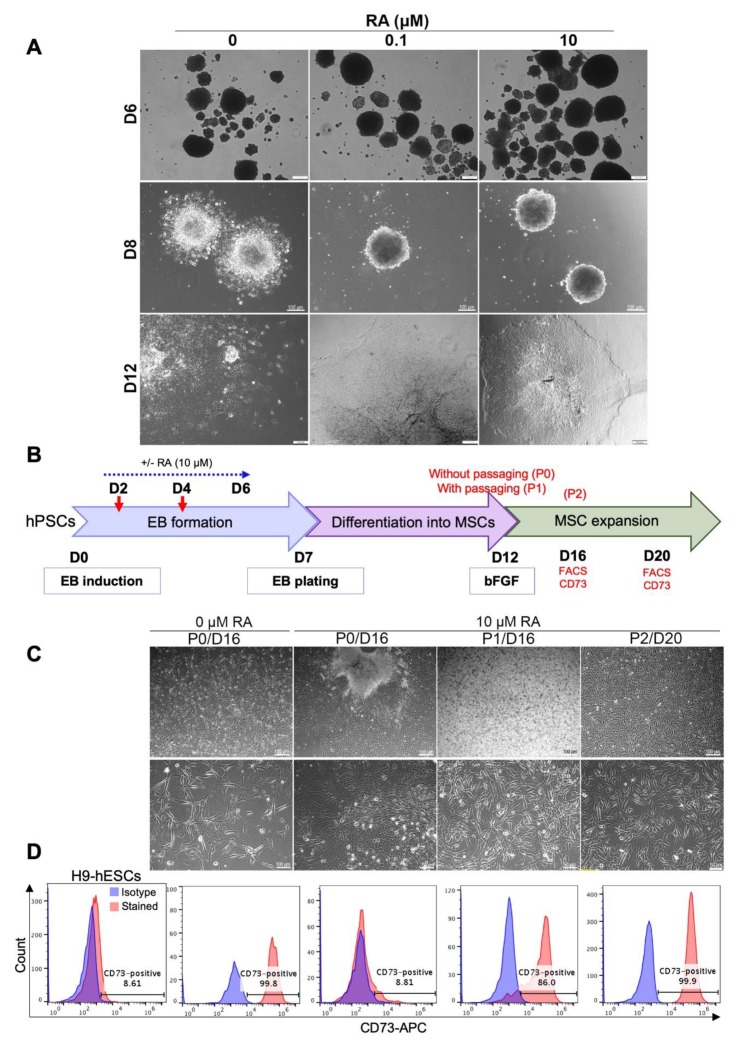
RA-mediated delay in EB differentiation can be reversed by cell dissociation. (**A**) Effect of RA treatment on EB morphology after plating on matrigel-coated dishes. H9 ESC-derived EBs treated or not with RA were plated on matrigel-coated dishes on day 7 of differentiation. Representative images were taken on days 6, 8 and 12 of differentiation. (**B**) Effect of dissociation and replating on the differentiation of RA-treated EBs into MSC-like cells. As indicated in the experimental scheme, H9 ESC-derived EBs treated or not with RA (10 µM condition) were dissociated (P1) or not (P0) on D12 of differentiation. A second dissociation was performed on D16 (P2). On D16 and D20, the cells were photographed to examine their morphology (**C**) and the expression of the MSC marker CD73 was analyzed by flow cytometry (**D**). Undifferentiated H9-hESCs were used as negative control. The images and histograms presented are representative from three independent experiments.

**Figure 3 cells-09-00710-f003:**
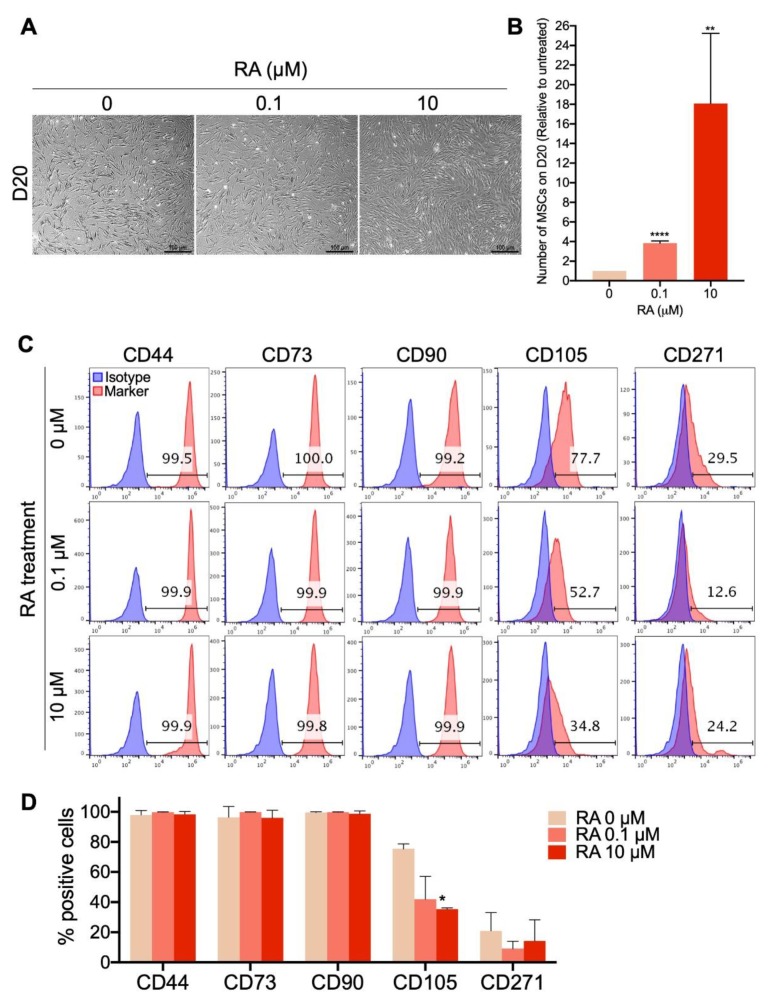
Short-term RA treatment enhances the yield of MSCs differentiated from hPSCs. H9-hESCs were differentiated into MSCs under the three different RA treatment conditions (0, 0.1 and 10 µM). MSC morphology, yield and phenotype were assessed on day 20 of differentiation. (**A**) Representative images of the differentiated fibroblast-like cells from at least 4 independent experiments are shown. (**B**) The numbers of MSC-like cells obtained under each of the RA treatment conditions were counted. The results presented in the bar graph are means ± SD for four independent experiments. ** *p* < 0.01 and **** *p* < 0.0001 versus RA untreated EB-derived MSC-like cells (0 µM). (**C**) Analysis of the expression of the MSC markers CD44, CD73, CD90, CD105 and CD271 in the generated MSC-like cells by flow cytometry. (**D**) The results presented in the bar graph are means ± SD for five independent experiments. * *p* < 0.05 versus MSCs derived from RA untreated EBs (0 µM).

**Figure 4 cells-09-00710-f004:**
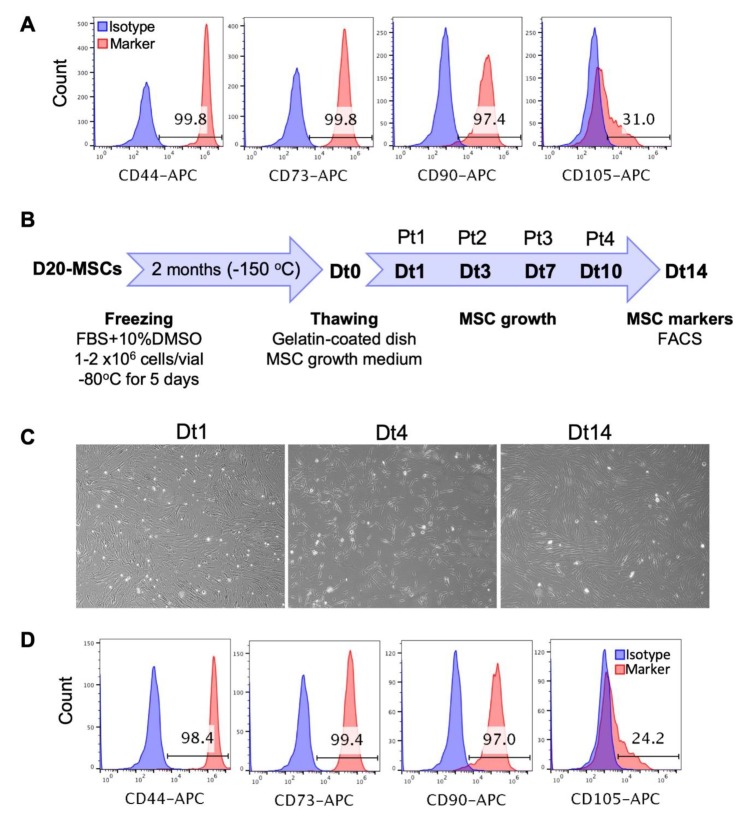
Long-term culture, freezing and thawing of the PSC-derived MSCs. (**A**) The MSCs derived from RA (10 µM)-treated H9 hESC-derived EBs were maintained and amplified in culture for 34 days (D50 of differentiation) and the expression of the MSC markers CD44, CD73, CD90 and CD105 was analyzed by flow cytometry. (**B**) The MSCs derived from RA (10 µM)-treated H9 hESC-derived EBs were frozen on D20 of differentiation as indicated in the experimental scheme. Two months later, these cells were thawed (day Dt0), cultured in the MSC growth medium on gelatin-coated dishes and passaged when they reached confluency on days 1, 3, 7 and 10 after thawing (Dt1, Dt3, Dt7 and Dt10). Pt1 to Pt4 indicate the number of passages after thawing. Cells were photographed on days 1, 4 and 14 after thawing (**C**) and the expression of MSC markers was analyzed on day 14 after thawing (Dt14) by flow cytometry (**D**).

**Figure 5 cells-09-00710-f005:**
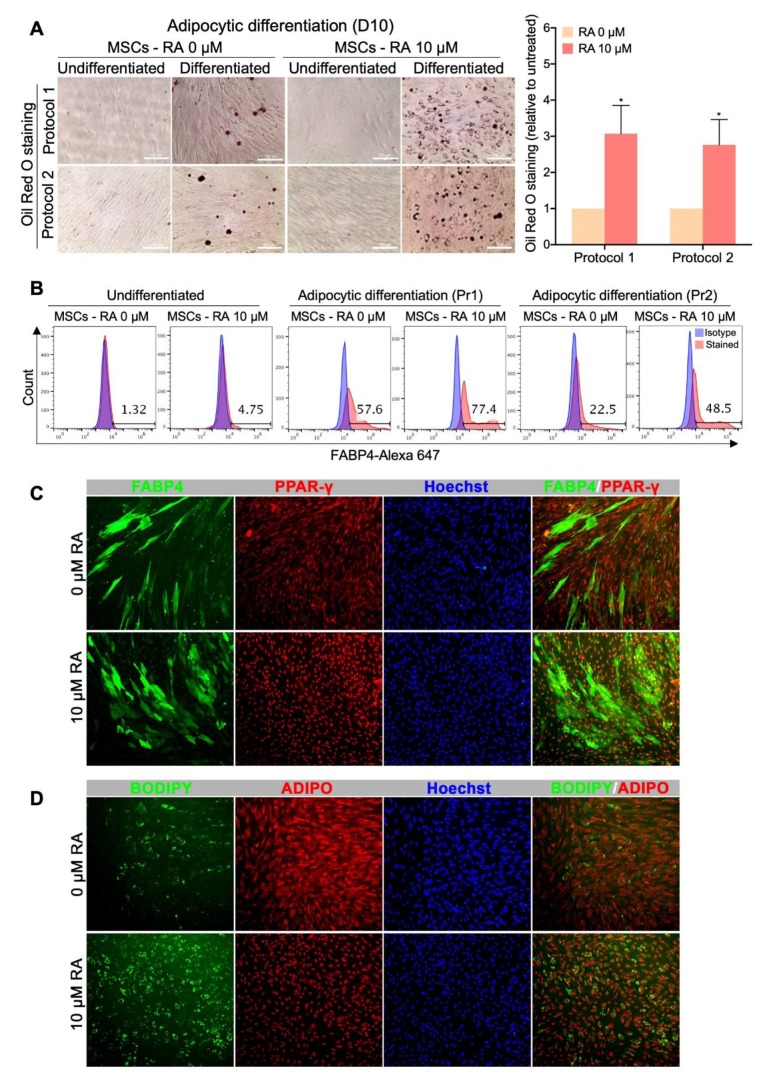
MSCs derived from RA-treated EBs display an enhanced differentiation potential into adipocytic lineage. MSCs obtained from H9-hESC-derived EBs treated or not with 10 µM RA were differentiated into adipocytes using two different previously reported protocols. (**A**) Representative Oil Red O staining images from at least 3 independent experiments are shown. Undifferentiated MSCs were used as control. The results of the Oil Red O staining quantification are presented in the bar graph as means ± SEM for three independent experiments conducted in duplicates or quadruplicates each. * *p* < 0.05 versus untreated EBs (0 µM). (**B**) Quantification of the expression of the adipogenesis marker FABP4 by flow cytometry. The histograms presented are representative of four independent experiments. Pr1 (protocol 1) and Pr2 (protocol 2) indicate the two different adipocytic differentiation protocols used. (**C**) Immunofluorescence images showing the expression of FABP4 and PPAR-γ in the adipocytes derived from the MSCs obtained from H9-hESC-derived EBs treated or not with 10 µM RA. (**D**) Immunofluorescence images showing lipid droplet staining by BODIPY and adiponectin (ADIPO) expression in the adipocytes derived from the MSCs obtained from H9-hESC-derived EBs treated or not with 10 µM RA.

**Figure 6 cells-09-00710-f006:**
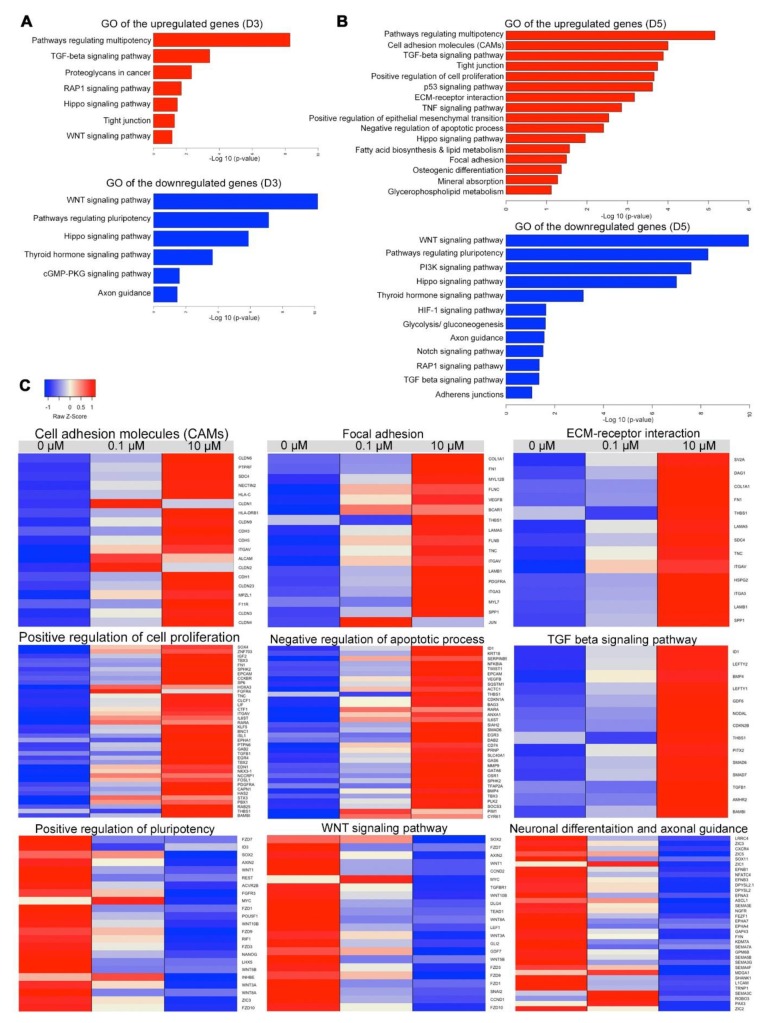
Differentially expressed genes in differentiated cells following RA treatment. Gene Ontology (GO) and Kyoto Encyclopedia of Genes and Genomes (KEGG) enrichment analysis of biological processes for upregulated and downregulated differentially expressed genes (DEGs) in H9-hESC-derived EBs treated with 10 μM RA or untreated EBs examined at day 3 (**A**) and day 5 (**B**). The enriched GO terms and KEGG pathways were plotted against –log10 (*p*-value). (**C**) Heatmaps showing DEGs in H9-hESC-derived EBs treated with 10 µM RA compared to those treated with 0.1 µM RA and untreated EBs at day 5 of differentiation. The upregulated genes were associated with cell adhesion molecules (CAMs), focal adhesion, ECM-receptor interaction, positive regulation of cell proliferation, negative upregulation of apoptotic process and TGF beta signaling pathway, while the downregulated genes were associated with positive regulation of pluripotency, WNT signaling pathway, neuronal differentiation and axonal guidance. The relative value for each gene is depicted by color intensity, with red indicating upregulated and blue indicating downregulated genes. See also Appendix A, Appendix A.

**Figure 7 cells-09-00710-f007:**
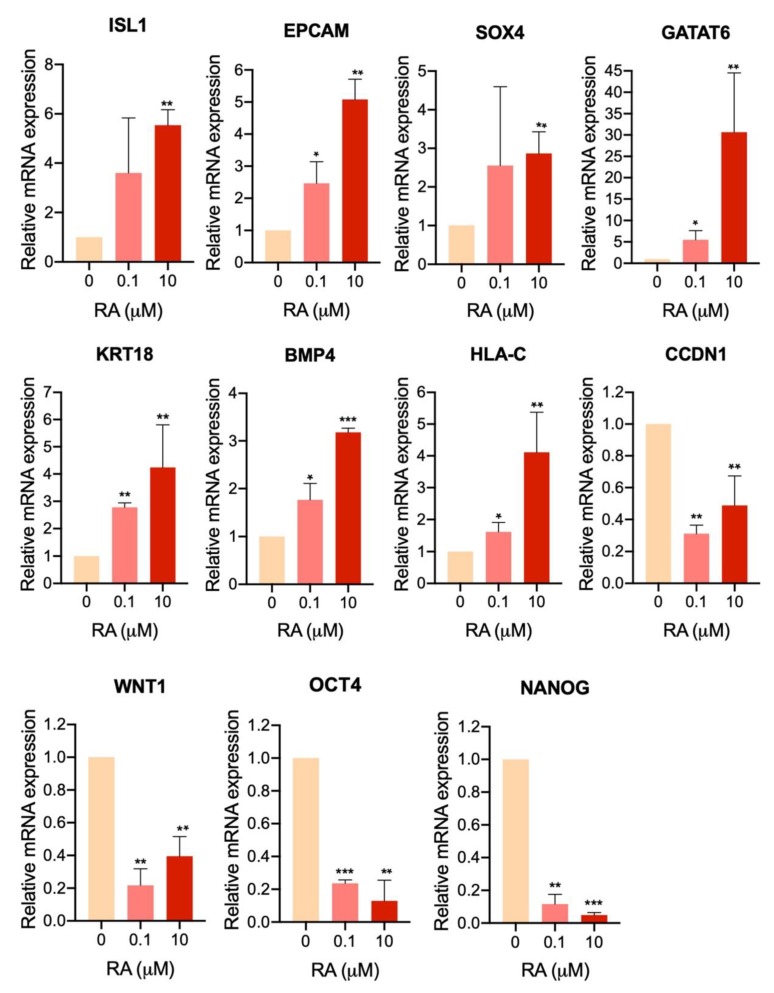
RNA-seq validation using quantitative PCR (qPCR). qPCR analysis for some differentially expressed genes (DEGs) in H9-hESC-derived EBs treated with 10 µM RA compared to those treated with 0.1 µM RA and untreated EBs at day 5 of differentiation. * *p* < 0.05, ** *p* < 0.01, *** *p* < 0.001.

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
