# Peer review of "Scalable Generation of Mesenchymal Stem Cells and Adipocytes from Human Pluripotent Stem Cells"

_cells, 2020, doi:10.3390/cells9030710_

Round 1
Reviewer 1 Report
The manuscript by Karam and colleagues described a modified method to enhance the production of MSCs from human pluripotent stem cells (hPSCs), either hESCs or hiPSCs, using All-trans retinoic acid (RA). The authors propose this method will allow an unlimited supply of MSCs, and subsequently adipocytes derived from the MSCs, for therapeutic application. The concept is of significant interest, and potentially important for the field of cell therapeutics, particularly if this method could provide effective and more homogeneous cells for use in patients. While the study is well done, from cell characterization as an MSC to RNAseq analysis, I do have some questions and concerns.
1) The authors assess the use of RA in both hESCs and hiPSCs. In all experiment it is important to define the origin of the MSCs, and not just state from hPSCs. For example, in the RNAseq evaluation, were the MSCs derived from embryonic stem cells or iPSC cells of human origin?
2) Moreover, in the RNA seq experiments, it is stated at least 2 biologic replicates were performed on different days. How are the statistics performed if in some samples there is only an n=2? Did the authors perform technical replications of the 2 biological samples to provide data for statistical evaluation? This needs clarification.
3) Did the authors perform any confirmatory evaluation of the RNAseq results on a subset of pathways felt to be important (proliferation, cell survival, adipogenesis), either by qRT-PCR on a different set of RNA samples, or importantly by assessing protein levels of important mediators? These data would better define the pathways that the authors are proposing as important.
4) The RA-derived MSCs display an enhanced ability to differentiate into adipocytes. Thus, this new method may well change the function of the MSC population. Since the mentioned goal of these experiments in the future is to provide an unlimited supply of MSCs for therapeutics, it is important to know whether the function of the hPSC-derived MSCs are altered by RA exposure, versus non-RA exposed MSCs, versus primary tissue hMSCs (such as from bone marrow, adipose tissue, or another source). To provide a functional comparison of the cells from the different groups would be of significant interest. For example, experiments assessing properties of the MSCs to promote anti-inflammation, immunosuppression, or anti-apoptosis in an injury model would be important – either from a cell-to-cell interaction or via a paracrine action – as many therapeutic clinical trials are treating inflammatory processes with associated tissue injury.
Author Response
Reviewer #1 Comments:
Q1. The authors assess the use of RA in both hESCs and hiPSCs. In all experiment it is important to define the origin of the MSCs, and not just state from hPSCs. For example, in the RNAseq evaluation, were the MSCs derived from embryonic stem cells or iPSC cells of human origin?
We thank the reviewer for his/her comments. Following the reviewer’s comments, we have defined the origin of the MSCs throughout the manuscript and in all Figures. H9 hESCs were used for the RNAseq evaluation. The origin of the cells is now clarified in the revised version of the manuscript for the RNAseq study as well as for the other experiments.
Q2. Moreover, in the RNA seq experiments, it is stated at least 2 biologic replicates were performed on different days. How are the statistics performed if in some samples there is only an n=2? Did the authors perform technical replications of the 2 biological samples to provide data for statistical evaluation? This needs clarification.
For the RNA-seq experiments, each sample had two biological replicates, and each one of these was sequenced in two lanes (that is each biological sample had two technical replicates). During analysis, the technical replicates were combined to provide the required depth of reads (>16 million uniquely mapped reads) for proper statistical analysis. The statistical testing for significance of differentially expressed genes is built into the program used, Cuffdiff2, which we set to consider the mapped reads from the two conditions of interest and used a rigorous statistical analysis to report genes that are differentially expressed. The detailed statistical approach is described in Trapnell et al. “Differential analysis of gene regulation at transcript resolution with RNA-seq. Nat Biotechnol. 2013 Jan;31(1):46-53.” According to the developers of Cuffdiff2: “The variability in fragment count for each gene across replicates is modeled. Then, the fragment count for each isoform is estimated in each replicate, along with a measure of uncertainty in this estimate arising from ambiguously mapped reads. The algorithm combines estimates of uncertainty and cross-replicate variability under a beta negative binomial model of fragment count variability to estimate count variances for each transcript in each library. These variance estimates are used during statistical testing to report significantly differentially expressed genes and transcripts.”
Q3. Did the authors perform any confirmatory evaluation of the RNAseq results on a subset of pathways felt to be important (proliferation, cell survival, adipogenesis), either by qRT-PCR on a different set of RNA samples, or importantly by assessing protein levels of important mediators? These data would better define the pathways that the authors are proposing as important.
Following the reviewer’s comment, we have validation experiments using real time PCR. We checked several genes and all the results were consistent with the RNA-seq data. The genes examined were ISL1, EPCAM, SOX4, GATA6, KRT18, BMP4, GLA-C, CCDN1, OWNT1, Oct4, and NANOG. We have added these details in the manuscript in a new Figure 7 and in the text (Page 18) as well as Materials and Methods.
Q4. The RA-derived MSCs display an enhanced ability to differentiate into adipocytes. Thus, this new method may well change the function of the MSC population. Since the mentioned goal of these experiments in the future is to provide an unlimited supply of MSCs for therapeutics, it is important to know whether the function of the hPSC-derived MSCs are altered by RA exposure, versus non-RA exposed MSCs, versus primary tissue hMSCs (such as from bone marrow, adipose tissue, or another source). To provide a functional comparison of the cells from the different groups would be of significant interest. For example, experiments assessing properties of the MSCs to promote anti-inflammation, immunosuppression, or anti-apoptosis in an injury model would be important – either from a cell-to-cell interaction or via a paracrine action – as many therapeutic clinical trials are treating inflammatory processes with associated tissue injury.
We agree with the reviewer that it would be of significant interest to provide a functional comparison of RA MSCs versus non-RA MSCs versus primary tissue MSCs. However, as indicated in the manuscript, the main purpose of the current study is to obtain MSCs that allow efficient generation of adipocytes to study adipogenesis, obesity and related diseases. Surprisingly, we also observed that the MSCs obtained from 10 µM RA-treated EBs not only exerted an increased differentiation potential into adipocytes but also were generated in dramatically increased amounts as compared to the RA-untreated control. The 10 µM RA-derived MSCs meet all criteria defining MSCs, including adherence to plastic, fibroblast-like morphology, surface expression profile and lineage commitment. The ability of the RA-derived MSCs to differentiated into chondrocytes and osteocytes suggests that their functions might be maintained even if their ability to differentiate into adipocytes is enhanced.
The analysis of the immunomodulatory properties of the RA-derived MSCs requires the application for specific ethical approvals to obtain and use primary tissue hMSCs, immune cells as well as a tissue injury model and would consist of a different study than the current one. However, from RNA sequencing data obtained from the current study, we could observe some promising hints since both RA and non-RA MSCs express (at basal unstimulated state) detectable to significant mRNA levels of some factors that mediate MSC immunomodulatory and paracrine functions (Jiang and Xu 2019 Cell proliferation). Those include TB4, galactin-1, TGF- β1, VEGFA, VEGFB, CCL2, PDL1, PDL2, BMP4, IL-6, HGF, TSG6, IL-10, IDO1 and IDO2, while the expression of other factors such as FasL and HLA-G was not detected in unstimulated state (Table 1 below, genes in black). The immunosuppressive activities of MSCs are primarily stimulated by pro-inflammatory cytokines such as IFN-γ and TNF-α. The expressions of the genes encoding the receptors for these cytokines were significantly detected in both RA and non-RA MSCs (IFNGR1 and TNFRSF1A, respectively) (Table S6 below, genes in blue), suggesting that RA MSCs would be responsive to pro-inflammatory factors that would induce their immuno-modulatory functions. These preliminary observations suggest that RA MSC more likely exert immunomodulatory and paracrine functions, which still needs to be confirmed. This point was discussed in the revised manuscript (Discussion section, pages 22 & 23, marked in red colour).
Table S6: RNA sequencing data for MSCs obtained from H9 hESCs without (Ctrl4) or with (T6) 10 µM RA treatment. Genes encoding factors that mediate the immunoregulatory and paracrine functions of MSCs are in black. Genes encoding receptors for pro-inflammatory factors that are involved in MSC activation are in blue.

Reviewer 2 Report
In this work, the authors present a method for the scalable generation of mesenchymal stem cells (MSCs) from hPSC (hESC/hiPSCs) and adipocytes differentiated from MSCs. With short treatment of hPSC-derived EBs with a high concentration of All-trans-retinoic acid (RA), cell proliferation, survival, self-renewal and differentiation were highly improved. The work provides a useful method to obtain MSCs and adipocytes. There are some questions and comments:
The major concern about this work is that iPSC’s data were only shown in figure S1 to demonstrate the RA treatment can promote the differentiation of iPSCs into EBs and MSCs. But there is no result in the manuscript whether RA treated iPSCs can differentiation into adipocytes and other cell types. As only hESC’s results were found in the manuscript, it is difficult to say that RA works on iPSCs’ adipogenic differentiation. Briefly describe the method for hiPSC generation. Can RA at a concentration more than 10 uM promote the differentiation to MSC as well as the adipogenic differentiation? The t-test was not suitable for Figure 1B, even the authors only compared the treatment group with a control group. If there are more than two groups in one figure, a one way ANOVA is necessary. Then you can use the Dunnet test to reveal the difference between the treatment and control groups. Line 383, adipocytes differentiation is Figure 6A or 5A? I suggest removing Table 1 to the supplementary files to increase the readability of the paper.Author Response
Q1. The major concern about this work is that iPSC’s data were only shown in figure S1 to demonstrate the RA treatment can promote the differentiation of iPSCs into EBs and MSCs. But there is no result in the manuscript whether RA treated iPSCs can differentiation into adipocytes and other cell types. As only hESC’s results were found in the manuscript, it is difficult to say that RA works on iPSCs’ adipogenic differentiation.
Following the reviewer’s comment, we perform new experiments to differentiate the MSCs generated from hiPSCs into adipocytes. Our results showed that MSCs obtained from hiPSC-derived EBs treated with 10 µM RA could efficiently differentiate into large number of adipocytes. The adipocytes expressed key adipocytic markers, such as FABP4, BODIPY, and adiponectin. These results have been added to text in the manuscript and we added a new Figure (Supplementary Fig. S5).
Q2. Briefly describe the method for hiPSC generation.
We extensively characterized the iPSC lines generated in this study. During the reviewing process, our manuscript, which includes all the generation and characterization details has been accepted and published online. Therefore, we cited the reference in the Materials and Methods (Ali et al. Stem Cells Dev, 2020: https://www.liebertpub.com/doi/full/10.1089/scd.2019.0150?url_ver=Z39.88-2003&rfr_id=ori:rid:crossref.org&rfr_dat=cr_pub%3dpubmed )
Q3. Can RA at a concentration more than 10 uM promote the differentiation to MSC as well as the adipogenic differentiation?
Following the reviewer’s comment, we have performed a new experiment to examine the effect of higher concentrations of RA on MSC and adipogenic differentiation. In the new experiment, we used 10 uM, 20 uM, 50 uM, and 100 uM. Interestingly, we found that the number of EBs was significantly reduced with high concentrations. We added these details in the manuscript in the text (pages 6 & 7) and added a new Figure (Supplementary Figure S2).
Q4. The t-test was not suitable for Figure 1B, even the authors only compared the treatment group with a control group. If there are more than two groups in one figure, a one way ANOVA is necessary. They you can use the Dunnet test to reveal the difference between the treatment and control groups.
The one-way ANOVA followed by Dunnett’s test was performed. Although the p values were increased compared to the t-test, the differences between the treatment and control groups remained significant. This was updated in the manuscript’s “Material and Methods” section (Statistical analysis) and in Figure 1B panel and legend.
Q5. Line 383, adipocytes differentiation is Figure 6A or 5A?
It is 5A. We corrected it in the manuscript
Q6. I suggest removing Table 1 to the supplementary files to increase the readability of the paper.
Following the reviewer’s comment, we deleted Table 1 from the text and added it as a Supplementary Table 3.

Round 2
Reviewer 1 Report
The authors have answered the majority of my questions / concerns. Although, providing some type of functional assessment of the RA-derived MSCs compared with non-RA exposed or primary cells would have strengthened the impact of the manuscript.